# Different Therapeutic Response to Anti-TNF Drugs in Patients with Axial Spondyloarthritis Depending on Their Clinical Profile: An Unsupervised Cluster Analysis

**DOI:** 10.3390/jcm13071855

**Published:** 2024-03-23

**Authors:** Carmen Priego-Pérez, María Ángeles Puche-Larrubia, Lourdes Ladehesa-Pineda, Jerusalem Calvo-Guitérrez, Rafaela Ortega-Castro, Alejandro Escudero-Contreras, Nuria Barbarroja, Eduardo Collantes-Estévez, Clementina López-Medina

**Affiliations:** 1Medical and Surgical Sciences Department, University of Cordoba, 14071 Cordoba, Spain; carmenpriegomedicina@gmail.com (C.P.-P.); lourdesladehesapineda@gmail.com (L.L.-P.); yeru83@hotmail.com (J.C.-G.); orcam84@hotmail.com (R.O.-C.); alexcudero2@gmail.com (A.E.-C.); barbarrojan@gmail.com (N.B.); educollantes@yahoo.es (E.C.-E.); clementinalopezmedina@gmail.com (C.L.-M.); 2Rheumatology Department, Reina Sofia University Hospital, 14004 Cordoba, Spain; 3Consolidated Group 05, Maimonides Biomedical Research Institute of Cordoba (IMIBIC), 14004 Cordoba, Spain

**Keywords:** axial spondyloarthritis, anti-TNF, effectiveness

## Abstract

**Background**: The objectives were as follows: (a) to identify, among patients with axial spondyloarthritis (axSpA), “clusters” of patients based on the presence of peripheral and extra-musculoskeletal manifestations (EMMs) and (b) to compare the effectiveness of the first anti-TNF drugs across the different clusters after 6 months of follow-up. **Methods:** An observational and retrospective study of 90 axSpA patients naïve to bDMARDs was conducted. An unsupervised cluster analysis using the “k-means” technique was performed using variables of peripheral and EMMs. Baseline clinical and sociodemographic characteristics were evaluated, and the response to anti-TNF treatment (considering responders as those with an improvement ≥1.1 for the Ankylosing Spondylitis Disease Activity Score (ASDAS) or ≥2.0 for the Bath Ankylosing Spondylitis Disease Activity Index (BASDAI)) was compared across the clusters after 6 months of follow-up. **Results:** Two clusters were identified: cluster 1 (*n* = 14), with a higher prevalence of peripheral manifestations, inflammatory bowel disease (IBD), and HLA-B27-positive status, and a lower prevalence of uveitis in comparison with cluster 2 (*n* = 76). Patients from cluster 1 experienced a more pronounced absolute improvement in ASDAS and BASDAI indices after 6 months. The percentage of responders after 6 months of follow-up was superior in cluster 1 compared to cluster 2 (85.7% vs. 48.7%, *p* = 0.011). **Conclusion:** This study suggests the existence of two clinical profiles in axSpA patients according to the peripheral and EMMs, with higher rates of anti-TNF effectiveness after 6 months in those with a greater presence of peripheral features.

## 1. Introduction

Spondyloarthritis (SpA) encompasses a highly heterogeneous group of rheumatic conditions characterized by inflammation of the axial skeleton, peripheral joints, and extra-articular manifestations (EMMs) like psoriasis, uveitis, and inflammatory bowel disease (IBD), and a strong association with the HLA-B27 antigen. Classically, SpA patients have been classified into various subgroups based on whether they present peripheral and/or EMMs such as ankylosing spondylitis (AS), psoriatic arthritis (PsA), arthritis/spondylitis associated with IBD, and reactive arthritis [1,2]. The 2009 ASAS (Ankylosing Spondylitis Assessment Study) criteria classifies these patients according to their clinical presentation as patients with predominantly axial SpA (axSpA) or predominantly peripheral SpA (pSpA) [3]. However, in patients with axSpA, peripheral manifestations are very common, with around 30–50% of axSpA patients presenting with concomitant peripheral involvement (i.e., arthritis, enthesitis, or dactylitis) [4]. Moreover, peripheral and EMMs (psoriasis, IBD, and uveitis) play a crucial role in diagnosing patients, as they are often the first presenting symptom [5].

Patients with axSpA are initially treated with nonsteroidal anti-inflammatory drugs (NSAIDs), but when these do not control the disease, they are treated with biological disease-modifying antirheumatic drugs (bDMARDs), of which the most frequent and first-line treatment options are the anti-tumor necrosis factors (anti-TNF) [6]. Randomized controlled trials have demonstrated the effectiveness of anti-TNF in treating axSpA [7]. However, discontinuation of treatment due to a lack of efficacy is common, while remission is rarely the reason for withdrawal [8]. Various studies have searched for predictors of anti-TNF response and adherence in axSpA patients [9,10,11]. Nevertheless, it remains unclear whether anti-TNF effectiveness changes according to their clinical profile, despite suggestive findings from some studies.

It has been observed that the presence of peripheral manifestations can influence the response to biological therapy [12]. However, there have been few studies on the response to biological therapy based on the patient’s clinical profile. The hypothesis of this study is that the clinical profile of patients (determined through unsupervised cluster analysis) is associated with a different response to biological therapy. 

The objectives of this study were as follows: a) to identify, in patients with axSpA, “clusters” or patient profiles based on the presence of peripheral and EMMs, and b) to compare the effectiveness of anti-TNF across the different clusters after 6 months of follow-up. 

## 2. Methods

### 2.1. Design and Patients

This is an observational, longitudinal, and retrospective study conducted on 90 patients with axSpA who were treated at the Rheumatology Department of the Reina Sofía Hospital in Córdoba (Spain). Patients were diagnosed by a rheumatologist as having axSpA and classified according to the 2009 ASAS classification criteria [3]. Patients were consecutively included following these inclusion criteria: patients over 18 years old, with a diagnosis of axSpA, and naïve to bDMARDs who initiated treatment with first-line anti-TNF drugs based on their clinical condition. Patients with other concomitant rheumatic diseases or those who started a bDMARD different than anti-TNF were excluded. Only patients who started biological treatment from 2014 onwards were selected for the study, since this was the date in which clinical records were digitized in our center.

Two visits separated by 6 months were recorded: the baseline visit in which patients initiated the anti-TNF and 6 months later after the initiation. 

The study was approved by the ethics committee of our Reina Sofia University Hospital on 23 February 2023. The study was subject to the rules of good clinical practice and at all times complied with the ethical precepts contained in the Declaration of Helsinki. All patients gave written consent for participation.

### 2.2. Collected Variables

Clinical and analytical variables derived from routine clinical practice were collected from the electronic medical records of the patients. The collected variables were the following:
-Sociodemographic data: sex, age, smoking status, and body mass index (BMI).-Clinical characteristics and SpA features: age of onset of axSpA, the initial symptom of low back pain, disease duration (years between symptom onset and the study visit of anti-TNF initiation), diagnostic delay (years between symptom onset and axSpA diagnosis), family history of SpA, and HLA-B27 antigen status. Peripheral (i.e., arthritis, enthesitis, dactylitis) and EMMs (i.e., uveitis, psoriasis, IBD) at any time during the course of the disease were collected.-Patient-reported outcomes (PROs): To measure disease activity indices and determine if the patient was a responder or not, the following data were collected at baseline (i.e., the day of the anti-TNF initiation) and at the 6-month follow-up visit: the Bath Ankylosing Spondylitis Disease Activity Index (BASDAI) [13], the patient’s global visual analog scale (global VAS), the patient’s medical visual analog scale (medical VAS) the patient’s total visual analog scale (total VAS), and the Ankylosing Spondylitis Disease Activity Score (ASDAS) [14] were collected for all patients to assess disease activity. The Bath Ankylosing Spondylitis Functional Index (BASFI) was used to evaluate function in these patients [15]. Finally, the C-reactive protein (CRP, mg/dL) and the erythrocyte sedimentation rate (ESR) were collected.Based on these data, at the 6-month follow-up, patients were classified into a new dichotomous variable (responders and non-responders) according to the decrease in disease activity indices following the ASAS/EULAR 2022 recommendations (considering an improvement ≥1.1 for the ASDAS index or ≥2.0 for the BASDAI index as a responder) [6].-Past and current treatment: Data on previous or concomitant treatments were collected, including nonsteroidal anti-inflammatory drugs (NSAIDs) and conventional synthetic disease-modifying antirheumatic drugs (csDMARDs) such as sulfasalazine, methotrexate, leflunomide, or corticosteroids. 

### 2.3. Statistical Analysis

Descriptive data were presented as means and standard deviations (SD) for quantitative variables and absolute and relative frequencies for qualitative variables.

Firstly, a cluster analysis was conducted to identify groups of patients according to their clinical profile. Variables used for clustering were peripheral manifestations (arthritis, enthesitis, and dactylitis), extra-articular manifestations (psoriasis, IBD, and uveitis), and HLA-B27 antigen. Clustering was conducted using an iterative partitioning k-means technique, and the optimal number of clusters was estimated using the “NbClust” package, which proposed the best clustering scheme from 30 indices [16].

Next, to evaluate the characteristics of the clusters, clinical features such as peripheral and extra-articular manifestations and HLA-B27 positivity were compared between the two groups using the chi-square test or Fisher’s exact test for qualitative variables. Then, other sociodemographic characteristics, SpA features, and treatments were compared across clusters using the Student’s *t*-test or Mann–Whitney U test for quantitative variables and the chi-square test or Fisher’s exact test for qualitative variables.

The mean changes after 6 months of follow-up on disease activity were compared across clusters. The mean changes of ASDAS, BASDAI, global VAS, and CRP were compared between clusters using the Student’s *t*-test or Mann–Whitney U test.

Finally, patients were classified into a new dichotomous variable, responders and non-responders at 6 months, according to the ASAS/EULAR 2022 recommendations (considering an improvement of ≥1.1 for the ASDAS or ≥2.0 for the BASDAI as responders) [6]. The percentage of responder patients was compared between the two clusters using the chi-square test or Fisher’s exact test.

All analyses were bilateral, and a *p*-value < 0.05 was considered significant. Data were collected, processed, and analyzed using IBM SPSS Statistics v.25 (SPSS, Inc., Chicago, IL, USA) and RStudio 1.4.1106.

## 3. Results

A total of 90 biologic-naïve patients with axSpA were consecutively included in the study, all with the required data available regarding clinical features, and baseline and 6-month PROs. In the overall population, 65.6% were men, with a mean age of 42.5 (11.8) years (Table 1). The mean disease duration was 11.9 (10.7) years, and the mean delay in diagnosis was 7.5 (9.2) years. A total of thirty-four (38.2%) patients were taking sulfasalazine, fourteen (16.1%) were taking methotrexate, and only one (1.1%) patient was taking leflunomide.

### 3.1. Cluster Identification according to the Peripheral and Extra-Musculoskeletal Manifestations

Cluster analysis revealed the existence of two differentiated clinical profiles or clusters: cluster 1 with 14 (15.6%) patients and cluster 2 with 76 (84.4%) patients (Figure 1). 

Table 1 shows the values of the variables used for the cluster analysis in all patients and the difference between both clusters. Overall, the comparison between the two clusters demonstrated that cluster 1 presented a significantly higher prevalence of peripheral manifestations in comparison with cluster 2, such as enthesitis (78.6% vs. 0.0%, respectively), arthritis (57.1% vs. 13.2%, respectively), and dactylitis (42.9% vs. 3.9%, respectively). In addition, IBD was more frequent in cluster 1 (35.7% vs. 5.3%) while uveitis was more frequent in cluster 2 (0% vs. 25.5%) (Table 1). Based on these characteristics, from now on we will we name cluster 1 as the “mixed phenotype”, while cluster 2 will be named as the “predominantly axial phenotype”.

### 3.2. Comparison of the Clusters

The comparison of sociodemographic and SpA-related characteristics (other features apart from those to identify the groups) between the two clusters is presented in Table 2. Patients in cluster 1 (mixed phenotype) had similar sociodemographic characteristics (age, sex, and BMI) to those in cluster 2 (predominantly axial phenotype) (Table 2). However, patients in cluster 2 had a significantly higher percentage of smokers (8.3% vs. 56.5% for clusters 1 and 2, respectively) (*p* = 0.002). In addition, disease duration was significantly longer in patients from cluster 2 (6.7 vs. 12.9 years for cluster 1 and 2, respectively) (*p* = 0.007), as was diagnosis delay (3.7 vs. 8.2 years for cluster 1 and 2, respectively) (*p* = 0.003).

More than 90% of patients in both clusters were taking NSAIDs, with no differences. Regarding disease-modifying antirheumatic drugs (DMARDs), a significantly higher percentage of patients in the mixed phenotype cluster were taking methotrexate (50.0% vs. 9.6% for clusters 1 and 2, respectively) (*p* < 0.001).

### 3.3. Comparison of Anti-TNF Effectiveness between the Two Clusters after 6 Months of Treatment

Table 3 shows that cluster 1 (mixed phenotype) showed a more pronounced improvement in ASDAS and BASDAI compared to cluster 2 (predominantly axial phenotype) after 6 months of follow-up (change in ASDAS −2.7 (1.5) vs. −1.6 (1.2), *p* = 0.029; change in BASDAI −4.1 (0.6) vs. −1.8 (0.3), *p* = 0.003) (Figure 2). Although the improvement in global VAS and CRP were higher for cluster 1, these differences were not significant.

Finally, the percentage of patients responding to anti-TNF (improvement ≥1.1 ASDAS or ≥2.0 BASDAI) after 6 months of follow-up was significantly higher for cluster 1, with 85.7% of respondents in cluster 1 vs. 48.7% in cluster 2 (*p* = 0.011) (Table 3 and Figure 2). There was also a significant improvement ≥2.0 in the BASDAI in cluster 1 (78.6% vs. 35.5% *p* = 0.003), but the improvement ≥1.1 in the ASDAS was not significant (35.7% vs. 27.6% *p* = 0.374).

## 4. Discussion

This study suggests the presence of two clinical profiles or “clusters” of patients with axSpA in clinical practice: a phenotype with a high prevalence of peripheral and EMMs (mixed phenotype) and a group of patients with a very low prevalence of peripheral and EMMs (predominantly axial phenotype). Interestingly, we found that patients from the former group appear to have a better response to anti-TNF compared to the predominantly axial phenotype after 6 months of follow-up. 

The results on the phenotypes are in line with previous studies conducted in the whole spectrum of SpA, in which peripheral features were found in both axial and peripheral phenotypes, with quantitative differences rather than qualitative [17]. In our analysis, we focused only on patients with axSpA, and we found a group with a higher prevalence of peripheral symptoms (cluster 1). Interestingly, our results showed that cluster 1, or the mixed phenotype, had a higher prevalence of HLA-B27 positivity than cluster 2, or the predominantly axial phenotype. This could be contradictory with the previous literature [18] showing that patients with peripheral symptoms are more frequently HLA-B27 negatives. However, this finding could be explained by several reasons. First, HLA-B27-positive patients with axial involvement and peripheral arthritis are more prone to be diagnosed as axSpA instead of PsA. Thus, the presence of HLA-B27 may lead the rheumatologist to establish a diagnosis of axSpA. Additionally, cluster 2 consisted of patients with axSpA with a high prevalence of psoriasis and uveitis, so many patients were diagnosed on extra-musculoskeletal manifestations rather than by HLA-B27. Another possible explanation could be the small sample size (90, with only 14 in cluster 1) which makes it difficult to extrapolate the findings.

In this study, we found that patients from cluster 1 (the mixed phenotype) appear to have a better response to anti-TNF compared to the predominantly axial phenotype after 6 months of follow-up. These findings seem to be in disagreement with previous studies [12] which showed that peripheral involvement was independently associated with persistently high disease activity. One possible explanation for this finding could be that the mixed phenotype may have a higher burden of overall inflammation and pain not only in the axial skeleton but also in peripheral joints, and therefore, the disease activity questionnaires (ASDAS, BASDAI) reflect a greater change. This could also explain why there were no significant changes in CRP. Another possible explanation for the lower response observed in cluster 2 may relate to its clinical presentation, which is predominantly axial but not exclusively so. Patients within cluster 2 exhibit concomitant peripheral and EMMs that could potentially impact treatment outcomes. An additional hypothesis for the lower response of cluster 2 could be a higher percentage of comorbidities in this group (although it is not significant, cluster 2 showed a greater use of medications for comorbid diseases). The study of comorbidities in patients with SpA has been of growing interest in recent years due to their potential impact on patient well-being and prognosis. In a prospective study, it has been shown that the increase in comorbidities in axSpA patients was associated with worse functional ability, higher disease activity, and worse mental and physical health compared to patients without comorbidities [11]. The possible higher prevalence of comorbidities in cluster 2 may have influenced the lower response ratio to anti-TNF in this group in comparison with cluster 1. Furthermore, cluster 2 had a higher percentage of smokers, longer disease duration, and diagnostic delay (although not significant), which could contribute to the worse response. Interestingly, one previous study demonstrated the association between HLA-B27 positivity with a better response to anti-TNF treatment in SpA patients [19]. This correlates with our findings, which showed a higher prevalence of HLA-B27 for cluster 1, as mentioned before, which was the cluster that responded the best.

Most patients experience the first symptoms in their second or third decade of life. The first line of pharmacologic defense is a nonsteroidal anti-inflammatory drug (NSAID) [6]. Among csDMARD, only sulfasalazine and methotrexate have demonstrated at least some therapeutic effect (i.e., limited effect on extra-axial arthritis but no effect on spinal inflammation) [20,21]. Therefore, it would be expected that patients within cluster 2, who exhibit predominantly axial symptoms, would not receive treatment with csDMARDs. Nevertheless, only methotrexate exhibits a significant difference in usage between the two clusters. It should be noted that the observed treatment pattern is because the clusters do not manifest as purely axial versus purely peripheral phenotypes, as mentioned above. Rather, cluster 2 reflects a predominantly axial phenotype with the presence of some patients exhibiting peripheral symptoms who require csDMARD treatment. Although these cases are limited, they are sufficient to impact the treatment response to anti-TNF results.

The variability in treatment response to anti-TNF therapy observed in our study highlights the need for a personalized approach to treatment. Different symptoms of the same disease can develop with different mechanisms and contribute to the view that the same drug can respond differently to these symptoms. This understanding can guide clinicians in personalizing treatment strategies to individual patients according to their main symptoms, thus optimizing therapeutic outcomes.

This study has some strengths and limitations. One limitation is the unicenter nature of the study, which may reduce the external validity of the results. A second limitation is the sample size, providing two groups of patients with one of them with only fourteen individuals. However, the classification of the clusters has been conducted using an unsupervised analysis, so the imbalance of number of individuals could be driven by the clinical characteristics. Another limitation is that the two clusters are not matched for confounders factors such as smoking, which might interfere with the results. One strength is indeed the use of this unsupervised analysis, which differentiates groups of patients according to their clinical characteristics without the influence of the investigator.

In summary, these results suggest the presence of two clinical profiles of patients with axSpA in clinical practice (a phenotype with a high prevalence of peripheral manifestations—mixed phenotype—and a predominantly axial phenotype). Patients with the mixed phenotype appear to have a better response to anti-TNF compared to the axial phenotype after 6 months of follow-up. These results confirm the significance of peripheral and extraarticular symptoms in the management of axSpA and indicate the need for further investigations to evaluate the role of other medical parameters (such as comorbidities or csDMARDs) in the differential response to treatment among these clinical subtypes. The identification of these clinical subtypes may facilitate tailored anti-TNF treatment strategies for patients with axSpA and ultimately improve clinical outcomes.

## Figures and Tables

**Figure 1 jcm-13-01855-f001:**
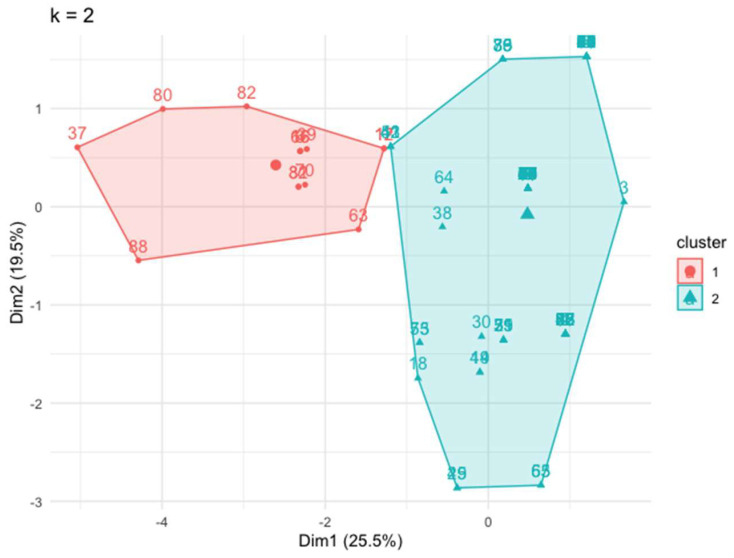
Two well-differentiated clusters generated by an unsupervised analysis using the “k means” technique. Dim: dimension.

**Figure 2 jcm-13-01855-f002:**
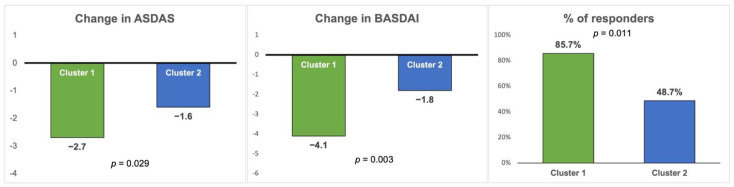
Comparison of changes in disease activity and therapeutic response between the two clusters at 6 months after the start of anti-TNF therapy. ASDAS: Ankylosing Spondylitis Disease Activity Score; BASDAI: Bath Ankylosing Spondylitis Disease Activity Index. Responders are defined as an improvement ≥1.1 ASDAS or ≥2.0 BASDAI.

**Table 1 jcm-13-01855-t001:** Comparison of clinical features between clusters.

	Total *n* = 90 *n* (%)	Cluster 1 *n* = 14 *n* (%)	Cluster 2 *n* = 76 *n* (%)	*p*-Value
HLA-B27-positive	71 (78.9%)	14 (100%)	57 (75.0%)	0.035
Arthritis	18 (20%)	8 (57.1%)	10 (13.2%)	<0.001
Enthesitis	11 (12.2%)	11 (78.6%)	0 (0.0%)	<0.001
Dactylitis	9 (10.0%)	6 (42.9%)	3 (3.9%)	<0.001
Uveitis	19 (21.1%)	0 (0.0%)	19 (25.5%)	0.035
Psoriasis	12 (13.3%)	1 (7.1%)	11 (14.5%)	0.683
IBD	9 (10.0%)	5 (35.7%)	4 (5.3%)	<0.001

IBD: inflammatory bowel disease.

**Table 2 jcm-13-01855-t002:** Comparison of sociodemographic and SpA-related characteristics between the two clusters.

	Total *n* = 90 *n* (%)	Cluster 1 *n* = 14 *n* (%)	Cluster 2 *n* = 76 *n* (%)	*p*-Value
Sex (male)	59 (65.6%)	10 (71.4%)	49 (64.5%)	0.764
Age, mean (SD)	42.5 (11.8)	38.0 (14.8)	43.4 (11.1)	0.058
Smoking	36 (48.6%)	1 (8.3%)	35 (56.5%)	0.002
BMI, mean (SD)	26.8 (5.3)	25.0 (5.2)	27.1 (5.3)	0.345
Obesity	14 (15.6%)	1 (7.1%)	13 (17.1%)	0.688
Disease duration, mean (SD)	11.9 (10.7)	6.7 (11.2)	12.9 (10.5)	0.007
Diagnosis delay, mean (SD)	7.5 (9.2)	3.7 (9.9)	8.2 (9.0)	0.003
Back pain as initial symptom	63 (79.7%)	7 (58.3%)	56 (83.6%)	0.060
Family history of SpA	35 (46.7%)	8 (72.7%)	27 (42.2%)	0.061
NSAIDs ever	84 (94.4%)	13 (92.9%)	71 (94.7%)	0.584
Sulfasalazine ever	34 (38.2%)	6 (42.9%)	28 (37.3%)	0.696
Methotrexate ever	14 (16.1%)	7 (50.0%)	7 (9.6%)	0.001
Leflunomide ever	1 (1.1%)	0 (0.0%)	1 (1.4%)	0.839
Corticosteroids ever	10 (11.2%)	1 (7.1%)	9 (12.0%)	0.509
Diabetes medication	3 (3.4%)	0 (0.0%)	3 (4.0%)	0.595
Hypertension medication	23 (25.8%)	2 (14.3%)	21 (28.0%)	0.235
Statins medication	11 (12.4%)	1 (7.1%)	10 (13.3%)	0.452

Chi-square or Fisher exact test. BMI: body mass index; NSAID: nonsteroidal anti-inflammatory drug; SD: standard deviation; SpA: spondyloarthritis.

**Table 3 jcm-13-01855-t003:** Comparison of changes in disease activity between the two clusters at 6 months after the start of anti-TNF therapy.

	Cluster 1 *n* = 14 Mean (SD)	Cluster 2 *n* = 76 Mean (SD)	*p*-Value
BASDAI (0–10)			
Baseline	5.9 (2.0)	5.6 (2.0)	
6 months	1.8 (1.4)	3.7 (2.5)	
Mean change	−4.1 (0.6)	−1.8 (0.3)	0.003
BASDAI question 1 (0–10)			
Baseline	6.2 (1.8)	6.3 (2.4)	
6 months	1.2 (1.1)	3.5 (2.4)	
Mean change	−5.0 (2.0)	−2.8 (2.6)	0.036
BASDAI question 2 (0–10)			
Baseline	6.4 (0.5)	7.1 (2.4)	
6 months	2.0 (2.9)	3.8 (3.1)	
Mean change	−4.4 (2.6)	−3.3 (3.4)	0.246
BASDAI question 3 (0–10)			
Baseline	6.4 (1.9)	4.0 (3.0)	
6 months	2.0 (2.9)	3.0 (2.9)	
Mean change	−4.4 (3.5)	−1.0 (2.9)	0.012
BASDAI question 4 (0–10)			
Baseline	6.8 (1.1)	5.2 (2.8)	
6 months	1.8 (2.5)	3.4 (3.0)	
Mean change	−5.0 (2.9)	−1.8 (3.1)	0.017
BASDAI question 5 (0–10)			
Baseline	4.8 (3.9)	6.1 (2.9)	
6 months	1.2 (1.8)	3.4 (3.1)	
Mean change	−3.6 (5.3)	−2.7 (3.5)	0.311
BASDAI question 6 (0–10)			
Baseline	3.2 (4.3)	5.3 (3.2)	
6 months	1.0 (1.4)	2.5 (2.9)	
Mean change	−2.2 (5.3)	−2.8 (3.0)	0.356
ASDAS			
Baseline	4.2 (1.4)	3.5 (1.0)	
6 months	1.5 (1.0)	1.9 (1.1)	
Mean change	−2.7 (1.5)	−1.6 (1.2)	0.029
Global VAS (0–100)			
Baseline	43.7 (30.8)	49.5 (31.1)	
6 months	13.5 (13.3)	20.3 (21.7)	
Mean change	−30.2 (30.3)	−29.2 (34.2)	0.887
CRP mg/L			
Baseline	27.7 (40.0)	11.2 (10.6)	
6 months	1.7 (2.2)	3.1 (5.1)	
Mean change	−25.9 (38.9)	−8.1 (10.5)	0.140
Improvement ≥ 1.1 ASDAS, *n* (%)	5 (35.7%)	21 (27.6%)	0.374
Improvement ≥ 2.0 BASDAI, *n* (%)	11 (78.6%)	27 (35.5%)	0.003
Improvement ≥ 1.1 ASDAS or ≥ 2.0 BASDAI, *n* (%)	12 (85.7%)	37 (48.7%)	0.011

ASDAS: Ankylosing Spondylitis Disease Activity Score; BASDAI: Bath Ankylosing Spondylitis Disease Activity Index; CRP: C-reactive protein; VAS; visual analogue scale.

## Data Availability

Data is available upon a reasonable request.

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
