# Peer review of "Different Therapeutic Response to Anti-TNF Drugs in Patients with Axial Spondyloarthritis Depending on Their Clinical Profile: An Unsupervised Cluster Analysis"

_jcm, 2024, doi:10.3390/jcm13071855_

Round 1

Reviewer 1 Report

Comments and Suggestions for Authors

Authors investigated anti-TNF response in 90 biologic-naïve patients with axSpA. They formed two clusters including Cluster 1 as “mixed phenotype”, and Cluster 2 as “pre-dominantly axial phenotype”.

They found that mixed phenotype had better anti-TNF responses in the sixth month compared to pre-dominant axial phenotype. Authors expressed the limitations of his work.

The study confirms clinical observations. In this respect, it does not make a new contribution to the literature. However, this work can provide additional evidence that different symptoms of the same disease can develop with different mechanisms and contribute to the view that the same drug can respond differently to these symptoms. I suggest that the Authors discuss this briefly.

Although the study does not contain very new data, it has features to help the clinician

Author Response

REVIEWER 1

Authors investigated anti-TNF response in 90 biologic-naïve patients with axSpA. They formed two clusters including Cluster 1 as “mixed phenotype”, and Cluster 2 as “pre-dominantly axial phenotype”.

They found that mixed phenotype had better anti-TNF responses in the sixth month compared to pre-dominant axial phenotype. Authors expressed the limitations of his work.

The study confirms clinical observations. In this respect, it does not make a new contribution to the literature. However, this work can provide additional evidence that different symptoms of the same disease can develop with different mechanisms and contribute to the view that the same drug can respond differently to these symptoms. I suggest that the Authors discuss this briefly.

Although the study does not contain very new data, it has features to help the clinician.

- Authors’ answer: Thank you for your suggestion.

- Author’s action: We have discussed this observation in the discussion section (5 th paragraph): “The variability in treatment response to anti-TNF therapy observed in our study highlights the need for a personalized approach to treatment. Different symptoms of the same disease can develop with different mechanisms and contribute to the view that the same drug can respond differently to these symptoms. This understanding can guide clinicians in personalizing treatment strategies to individual patients according to main symptoms, optimizing therapeutic outcomes.”

Reviewer 2 Report

Comments and Suggestions for Authors

Thank you for presenting this concise manuscript which is well-written. I have attached a number of comments for your consideration:

1. Discussion. "However, this finding could be explained by several reasons. First, patients HLA- 196 B27 positives with axial involvement and peripheral arthritis are more prone to be diag- 197 nosed as axSpA instead of PsA. Thus, the presence of HLA-B27 may lead to the rheuma- 198 tologist to stablish a diagnosis of axSpA. Another possible explanation could be the small 199 sample size (90, with only 14 in cluster 1) that avoid extrapolating the findings.".. The p value was not very significant. Could the cluster 2 (axial patients) with B27 negative disease have non-inflammatory back conditions causing stiffness, for example?

2. Discussion. "This study has some strengths and limitations. ". Another limitation is that your 2 clusters are not matched for important factors such as smoking, which might confound your results. I suggest to mention this in your discussion.

3. Table 3. Is it possible to break down the components of BASDAI which showed most improvement between clusters 1 and 2? This would allow you to identify if certain aspects of the disease are improving compared to others.

4. Reference 3. This is an editorial. Please cite the original article which may be: https://ard.bmj.com/content/70/1/25

Author Response

Thank you for presenting this concise manuscript which is well-written. I have attached a number of comments for your consideration:

  1. Discussion. "However, this finding could be explained by several reasons. First, patients HLA- 196 B27 positives with axial involvement and peripheral arthritis are more prone to be diag- 197 nosed as axSpA instead of PsA. Thus, the presence of HLA-B27 may lead to the rheuma- 198 tologist to stablish a diagnosis of axSpA. Another possible explanation could be the small 199 sample size (90, with only 14 in cluster 1) that avoid extrapolating the findings.".. The p value was not very significant. Could the cluster 2 (axial patients) with B27 negative disease have non-inflammatory back conditions causing stiffness, for example?

- Authors’ answer: Thank you for this comment. We would like to clarify that all patients in both clusters have a diagnosis of axial SpA. Our interpretation, apparat from that previously explained in the manuscript, is that cluster 2 consisted of patients with axial SpA with a high prevalence of psoriasis and uveitis, so many patients were diagnosed on extra-musculoskeletal manifestations rather than by HLA-B27.

- Author’s action: We have included this explanation in the discussion section (2 th paragraph): “Besides, cluster 2 consisted of patients with axSpA with a high prevalence of psoriasis and uveitis, so many patients were diagnosed on extra-musculoskeletal manifestations rather than by HLA-B27.”

  1. Discussion. "This study has some strengths and limitations. ". Another limitation is that your 2 clusters are not matched for important factors such as smoking, which might confound your results. I suggest to mention this in your discussion.

- Authors’ answer: Thank you for your recommendation.

- Author’s action: We have improved and expanded the limitations of the study in the discussion section (6 th paragraph): “Another limitation is that the two clusters are not matched for confounders factors such as smoking, which might interfere the results.”

  1. Table 3. Is it possible to break down the components of BASDAI which showed most improvement between clusters 1 and 2? This would allow you to identify if certain aspects of the disease are improving compared to others.

- Authors’ answer: Thank you for your suggestion.

- Author’s action: We have modified table 3 and divided the components of the BASDAI, observing that in questions 1 (fatige), 3 (joint pain/swelling) and 4 (areas of localized tenderness) there is greater improvement in Cluster 1 compared to Cluster 2.

  1. Reference 3. This is an editorial. Please cite the original article which may be: https://ard.bmj.com/content/70/1/25

- Authors’ answer: Thank you for pointing that out.

- Author’s action: We have corrected the erroneous reference:Rudwaleit M, van der Heijde D, Landewé R, et al (2011) The Assessment of SpondyloArthritis international Society classification criteria for peripheral spondyloarthritis and for spondyloarthritis in general. Annals of the Rheumatic Diseases 70:25-31”.